Food insecurity increases energetic efficiency, not food consumption: an exploratory study in European starlings

Bateson Melissa melissa.bateson@ncl.ac.uk 1
Andrews Clare 1
Dunn Jonathon 1
Egger Charlotte B.C.M. 1
Gray Francesca 1
Mchugh Molly 1
Nettle Daniel 2
1 Biosciences Institute/Centre for Behaviour and Evolution, Newcastle University , Newcastle upon Tyne , United Kingdom
2 Population Health Sciences Institute/Centre for Behaviour and Evolution, Newcastle University , Newcastle upon Tyne , United Kingdom
Curley James
Electronic publication date: 2021 May 28
Publication date: 2021
Volume: 9
Electronic Location ID: e11541
Received 2021 Mar 10; Accepted 2021 May 10
Copyright: ©2021 Bateson et al.
Copyright year: 2021
Copyright holder: Bateson et al.
License: This is an open access article distributed under the terms of the Creative Commons Attribution License, which permits unrestricted use, distribution, reproduction and adaptation in any medium and for any purpose provided that it is properly attributed. For attribution, the original author(s), title, publication source (PeerJ) and either DOI or URL of the article must be cited.
License URL: https://creativecommons.org/licenses/by/4.0/

Keywords: Food insecurity, Insurance hypothesis, Unpredictable food, Obesity, Overweight, Energy balance, Food consumption, Starvation risk, Starling

Funding: UK Biotechnology and Biological Sciences Research Council BB/J016446/1 European Research Council Advanced Grant to Daniel Nettle under the European Union’s Horizon 2020 research and innovation programme AdG 666669 Funding was provided by a UK Biotechnology and Biological Sciences Research Council project grant to Melissa Bateson and Daniel Nettle (BB/J016446/1) and a European Research Council Advanced Grant to Daniel Nettle under the European Union’s Horizon 2020 research and innovation programme (AdG 666669, COMSTAR). The funders had no role in study design, data collection and analysis, decision to publish, or preparation of the manuscript.

==============================
Food insecurity—defined as limited or unpredictable access to nutritionally adequate food—is associated with higher body mass in humans and birds. It is widely assumed that food insecurity-induced fattening is caused by increased food consumption, but there is little evidence supporting this in any species. We developed a novel technology for measuring foraging, food intake and body mass in small groups of aviary-housed European starlings (Sturnus vulgaris). Across four exploratory experiments, we demonstrate that birds responded to 1–2 weeks of food insecurity by increasing their body mass despite eating less. Food-insecure birds therefore increased their energetic efficiency, calculated as the body mass maintained per unit of food consumed. Mass gain was greater in birds that were lighter at baseline and in birds that faced greater competition for access to food. Whilst there was variation between experiments in mass gain and food consumption under food insecurity, energetic efficiency always increased. Bomb calorimetry of guano showed reduced energy density under food insecurity, suggesting that the energy assimilated from food increased. Behavioural observations of roosting showed inconsistent evidence for reduced physical activity under food insecurity. Increased energetic efficiency continued for 1–2 weeks after food security was reinstated, indicating an asymmetry in the speed of the response to food insecurity and the recovery from it. Future work to understand the mechanisms underlying food insecurity-induced mass gain should focus on the biological changes mediating increased energetic efficiency rather than increased energy consumption.

Introduction

The ‘food-insecurity-obesity paradox’ refers to the robust positive association found in humans living in Western developed countries between food insecurity—defined as limited or unpredictable access to nutritionally adequate food—and obesity (Dinour, Bergen & Yeh, 2007; Nettle, Andrews & Bateson, 2017). We have recently hypothesized that a functional explanation for this association may be found in a well-developed literature in behavioural ecology exploring how animals respond to temporally variable access to food (Nettle, Andrews & Bateson, 2017). The basic argument is that when access to food is limited and unpredictable, and there is therefore some probability of energetic shortfall, it is optimal to increase fat reserves in order to provide insurance against starvation (Lima, 1986; McNamara & Houston, 1990; Bednekoff & Houston, 1994; Higginson, McNamara & Houston, 2016; Nettle, Andrews & Bateson, 2017). However, while this functional argument is theoretically sound, from a mechanistic perspective it is currently unclear how either humans or birds achieve increases in body fat when access to food is limited and unpredictable (Anselme & Güntürkün, 2019; Kowaleski-Jones, Wen & Fan, 2019).

It is widely assumed that food-insecure individuals are fatter because they consume more during periods when food is available, leading to higher energy intake overall (Dietz, 1995; Brunstrom & Cheon, 2018; Anselme & Güntürkün, 2019). Specifically, Anselme and Güntürkün suggest that a stress response evoked by unpredictable food magnifies food-seeking motivation, leading to greater total food consumption and hence increased body mass (Anselme, Otto & Güntürkün, 2017; Anselme & Güntürkün, 2019). However, the evidence that food-insecure humans actually have higher energy intake is weak. While food-insecure participants eat more when given staged opportunities in the laboratory (Nettle et al., 2018; Stinson et al., 2018; Folwarczny et al., 2021), there is no evidence that they actually eat more in their everyday environment (Tarasuk & Beaton, 1999; Zizza, Duffy & Gerrior, 2008; Bergmans et al., 2018; Kowaleski-Jones, Wen & Fan, 2019; Nettle & Bateson, 2019). Furthermore, a longitudinal study found that increases in body mass index in pre-school girls whose households transitioned from food security to food insecurity were not associated with changes in either diet quality or total energy consumption (Jansen et al., 2017). Thus, it is currently unclear that the mass gain associated with food insecurity is caused by increased energy consumption. It is possible that food insecurity-induced mass gain could result from decreased energy expenditure, or increased assimilation of energy from food (see (Halsey, 2018) for a discussion of these mechanisms in relation to weight loss). However, in humans, it is difficult to measure food intake accurately in the home environment. Moreover, obese individuals systematically underreport their food consumption (Schoeller, 1995), meaning that estimates of consumption associated with food insecurity could be biased. One approach to resolving this issue is to study the response to food insecurity in animal experiments where food insecurity can be experimentally manipulated and food consumption accurately measured.

Here we report the results from a series of four experiments in which we tested the effects of 1–2 weeks of food insecurity on body mass and food intake in European starlings (Sturnus vulgaris). The starling is an omnivorous passerine bird that has been used as a model in laboratory studies of foraging behaviour and body mass (Asher & Bateson, 2008). Passerine birds are good species for studies of body mass, because of their rapid and accurate mass regulation (Halsey, 2018). Starlings thrive in temperate regions characterised by seasonal variation in food availability and respond rapidly to environmental changes that alter optimal fat reserves. Exposure to temporally variable and unpredictable food access causes increased fat deposition and body mass (Witter, Swaddle & Cuthill, 1995; Witter & Swaddle, 1997; Cuthill et al., 2000; Buchanan et al., 2003; Bauer et al., 2011). However, there are few data on how unpredictable food affects foraging motivation or energy intake, with the limited data available showing no change in food consumption when birds are exposed to unpredictable food availability, despite increases in body mass (Cuthill et al., 2000). Furthermore, there are some intriguing data from other passerine species (Paridae) reporting that under limited and unpredictable food, body mass can increase despite food consumption decreasing (Bednekoff & Krebs, 1995; Cornelius et al., 2017). These latter data suggest that the assumed causal link between increased food consumption and mass gain deserves closer scrutiny because there are alternative mechanisms that animals could use to adaptively alter body mass (Halsey, 2018). Therefore, our aim in the current study was to develop an ecologically-valid model of food insecurity in starlings and use this to test whether food insecurity-induced mass gain is caused by increased food consumption.

We conducted four experiments on small groups of starlings maintained in large aviaries (henceforth designated experiments 1–4; see Fig. 1 for a summary). Informed by analysis of human 24-hour food recall data from the US National Health and Nutrition Examination Survey, in which we showed that the strongest behavioural correlate of food insecurity is higher variance in the time intervals between food consumption events (Nettle & Bateson, 2019), we operationalised food insecurity as the existence of variable time intervals between periods of food availability. Additionally, because the questionnaires for measuring human food insecurity include items that probe anxiety about perceived future food availability (Saint Ville et al., 2019), we programmed the variance in the inter-food access interval such that it was unpredictable to the birds. Therefore, our operationalisation of food insecurity combined restricted access to food with variability and unpredictability in the intervals between periods of access. This was done intentionally to mimic the human experience of food insecurity and also the assumptions of the optimality models cited above. It was not our intention in the current study to differentiate between the effects of predictable and unpredictable variance in the timing of food, which have been studied elsewhere (Bateson & Kacelnik, 1997; Cuthill et al., 2000).

Figure 1 Experimental designs in experiments 1–4.

The second column indicates the method used to induce food insecurity (see text for details). The third column shows the number of aviaries in the experiment and the number of birds per aviary. The fourth column shows the number of social foraging system feeding stations (SFSs) per bird (see Fig. 2), providing an indication of the level of competition for food in the experiment (more birds per SFS indicates higher competition). The fifth column shows the order of treatments with each week comprising seven days: food-secure treatments are shown in blue (FS) and food-insecure treatments in red (FI). For treatments that lasted 2 weeks, the first week is designated a, and the second week b, but weeks a and b are identical and are pooled in all statistical analyses. In experiment 4, FIlow and FIhigh are treated as two separate treatments in statistical analysis.

The four experiments that we conducted used within-subjects designs, in which all subjects experienced all treatments: birds were exposed to a baseline week of food security (FS1), during which food was available immediately a bird sought access, followed by either one or two weeks of food insecurity (FI), during which food was both limited and unpredictable. In two of the four experiments (1 and 3), food security was reinstated for a final period (FS2) of one or two weeks following the period of food insecurity in order to study how rapidly the effects of food insecurity reversed.

In experiments 1–3, food insecurity was induced by withdrawing food access for a randomly chosen 12 out of 20 contiguous 20-minute periods starting two hours after dawn each day (designated ‘total food removal’ in Fig. 1). In experiment 1, this regimen resulted in a mean of 4.42 intervals per day without food, ranging from 20–120 min (mean ±sd = 54.19 ±36.22), and approximated the type of unpredictable access previously reported to induce mass gain in starlings (Cuthill et al., 2000). Intervals of similar duration were generated in experiments 2 and 3. In experiment 4, food insecurity was induced via an operant schedule in place from dawn until dusk (designated ‘probabilistic schedule’ in Fig. 1.), whereby the probability that a single peck to an illuminated key was reinforced with 10 s of food access was reduced from 1.0 under food security to 0.4 and later 0.2 under food insecurity (FIlow and FIhigh respectively). This probabilistic schedule would result in mean intervals of 20 ± 23.24 and 50 ± 53.66 s between periods of food access in FIlow and FIhigh respectively, assuming the birds pecked as soon as the key was illuminated. The schedule mimics an operant schedule used to study diurnal patterns in responses to decreased food availability in coal tits (Periparus ater) (Polo & Bautista, 2006). Thus, both methods for inducing food insecurity introduced variable intervals between periods of free access to food, albeit on different time scales. Our reasons for exploring the effects of the probabilistic schedule (experiment 4) in starlings for the first time were as follows. First, because the schedule does not induce long periods without food, it produces smoother and more consistent mass gain trajectories across the day that should facilitate comparisons between treatments (see Cuthill et al. (2000) for a discussion of the methodological problems inherent in total food removal methods). Second, using an operant schedule makes it possible to measure individual foraging motivation via key pecking data. Finally, the probabilistic schedule more closely embodies the assumptions of Anselme and Güntürkün’s mechanistic model of how unpredictable food causes mass gain (Anselme, Otto & Güntürkün, 2017; Anselme & Güntürkün, 2019), permitting a stronger test of this model than the total food removal method.

Under both methods for inducing food insecurity, it was theoretically possible for the birds to maintain or increase their total daily food consumption if they were motivated to do so. They could do this, either by eating more during the periods of the day when food was freely available (a total of 5 h spread across the day in experiments 1–3 and during the 10-second reinforcement periods in experiment 4), or by foraging for a greater proportion of the 14.5-hour day in experiment 4, in which birds could earn a theoretical maximum of 4.83 and 2.42 total hours of food access per day in treatments FIlow and FIhigh respectively.

In all experiments, a homogeneous complete diet was delivered from custom-built operant feeding stations (the social foraging system, SFS; Fig. 2) that additionally measured detailed individual mass data via integrated microchip-based identification and an electronic balance. Experiments were conducted in closed economy, meaning that birds received almost all of their daily food from the SFS (an additional four mealworms per bird were fed in all treatments during daily husbandry as a nutritional supplement). To measure effects of food insecurity on energy balance, we recorded the total amount of food consumed per day in each aviary. In a subset of experiments, we additionally measured foraging behaviour (key pecks; experiment 4), inactive roosting behaviour (experiments 1–3) and energy density of guano (faeces plus uric acid; experiments 1 and 3).

The four experiments described were preliminary studies conducted during the development and testing of the SFS. As such, the experiments were individually relatively small, with sample sizes of only six birds, and the methods varied somewhat (see Fig. 1 and Table S1). Therefore, the results presented below focus on meta-analysis of the four experiments to reveal any consistent effects of food insecurity that emerge despite the variation in methodology. Triangulation of this type is recognised as a strategy for increasing the robustness of scientific findings (Munafò & Smith, 2018; Bateson & Martin, 2021).

Materials & Methods

Subjects and basic husbandry

Subjects were 12 adult European starlings (Sturnus vulgaris) originally sourced from nest boxes in Northumberland, UK as chicks and hand-reared in the lab. The numbers, sexes and ages of birds used in each experiment are given in Table S1. The sample size of 6 birds used in each experiment was chosen based on a previous comparable experiment with starlings that reported an effect of unpredictable food on body mass (Cuthill et al., 2000). All of the birds had previous experience in a range of behavioural experiments. During the current experiments, the birds were group-housed in indoor aviaries. In experiments 1–3 the aviaries were rooms (215 × 340 × 220 cm) and in experiment 4, large walk-in cages (90 × 183 × 183 cm) with one cage per room. In all experiments, room temperature and humidity were maintained at ∼18 °C and ∼40% respectively. Each aviary was furnished with rope perches, wood chippings on the floor, a water bath, a drinker providing ad libitum clean water supplemented with vitamins and either one or two feeding stations (see below). The light cycle, number of birds per aviary and SFS stations per aviary for each experiment are given in Table S1. In all experiments, birds were maintained on a nutritionally complete diet of commercial poultry starter crumb (Special Diets Services Poultry Starter; henceforth ‘food’). Each bird was supplemented with 4 live mealworms daily. Daily husbandry took place at 1700 in experiments 1–3 and at 1100 in experiment 4.

Social foraging system

In all experiments, food was delivered from custom-built feeding stations (the social foraging system, SFS; Fig. 2). An SFS station comprised a retractable food hopper and an illuminable pecking key, both of which could only be accessed via a perch designed to accommodate a single bird (the pecking key was only used in experiment 4). The perch was mounted on a load cell that functioned as an electronic balance that measured to a resolution of 0.01 g. Beneath the perch there was an aerial tuned to read microchips (radio-frequency identification devices, RFIDs), that were mounted on leg rings of the starlings. Each aviary contained either one or two SFS stations depending on the experiment (Table S1). The SFS stations were connected to a computer in an adjacent room running Whisker experimental control software and additional custom-written software. This computer collected continuous data on the identity and masses of birds visiting each station and, in experiment 4, also controlled the operant schedule in place on each station.

Manipulations of food insecurity

In experiments 1–3 food security was created by maintaining the SFS hopper in the raised position, and thus available for foraging from dawn until dusk. Food insecurity was induced by withdrawing the hopper for 12 out of a possible 20 randomly chosen 20-min periods starting two hours after lights-on each day, and ending 1 h and 20 min prior to lights-off.

In experiment 4 the default position of the food hopper was lowered such that food was unavailable. Food security was created via a ratio schedule, whereby a single peck at the illuminated key on the SFS caused the key light to extinguish and the hopper to raise allowing 10-s access to food. At the end of the reinforcement, the hopper lowered and there was a 2-s inter-trial interval before the key re-illuminated signalling the start of the next available trial. The operant sessions began at 15 min after lights-on each day (0615) and ended 15 min before lights-off (2045). This schedule was designed to mimic a starling foraging by probing for hidden soil invertebrates. Food insecurity was induced by reducing the probability that a key peck was reinforced on any given trial first to 0.4 under FIlow, and second to 0.2 under FIhigh. Reducing the probability of reinforcement below one increased the variance in the inter-reinforcement interval. On unreinforced trials, a peck caused the key light to extinguish and there was a 10-s time out delay during which the hopper remained lowered followed by a 2-s inter-trial interval. Under food security, the maximum number of reinforcements available per day from the SFS was 4350 (assuming that birds pecked as soon as they key illuminated). Under food insecurity, the expected maximum number of reinforcements available per day dropped to 1740 and then 870 under FIlow and FIhigh respectively.

Procedure

All experiments began with birds being caught from their home aviary, manually weighed and equipped with two plastic leg rings each of which had a unique RFID microchip attached (birds wore two chips to guard against data loss in the event that one chip fell off, broke or was not read due to poor alignment with the SFS aerial). Birds were then released into experimental aviaries. Food was initially provided ad libitum by raising the hoppers of the SFS stations. Birds were initially encouraged to visit the SFS by placing mealworms around the stations and in the food hopper, but as soon as they were visiting the SFS readily these extra mealworms were withdrawn. The first experimental treatment of experiments 1–3 began once the birds were maintaining stable body masses on food obtained from the SFS.

Experiment 4 required additional training for the birds to learn that food could only be accessed by pecking the illuminated key on the SFS. We used an auto-shaping procedure whereby illumination of the pecking key signalled subsequent unconditional raising of the food hopper. Birds first learned a Pavlovian association between the lit key and food and spontaneously started performing appetitive pecks to the lit key. Pecks to the lit key were reinforced by immediate hopper raising, thus creating an instrumental association between key pecking and food. As soon as key pecking was established, hopper raising was made conditional on key pecking and the birds were moved to continuous foraging, whereby they earned all of their daily food by pecking at the lit key on the SFS. The first experimental treatment of experiment 4 began once the birds were maintaining stable body masses on food earned from the SFS.

The sequence and duration of the treatments experienced in each experiment is shown in Fig. 1. In all experiments, birds were maintained in closed economy, obtaining the majority of their daily food intake from the SFS. The diet was supplemented with four mealworms per bird given during daily husbandry and supplied in two spatially separated bowls to prevent one bird from monopolising the worms. The experiments ran seven days a week with no gap between treatments. The birds’ welfare was monitored daily via visual checks and using the body mass data obtained from the SFS balances; criteria for consulting a vet and considering euthanasia were either presence of puffed feathers and lethargy and/or a mass of <62 g. No bird became unwell or dangerously thin at any point during the study and no birds died or were euthanized. At the end of the final treatment of each experiment the birds were re-caught, manually weighed and returned to their home aviaries where they were retained for further studies. Experiments 1–3 were conducted consecutively on the same group of 6 male birds with a 24-day break in the home aviary between experiments 1 and 2 and an 18-day break between experiments 2 and 3.

Figure 2 A social foraging system (SFS) station.

Each station was designed to accommodate a single starling at any one time. The smooth pyramids prevented birds from perching on the top of the SFS or adjacent to the short perch on the balance. The food hopper was operated with a stepper motor and could be switched between a raised, accessible position and a lowered, inaccessible position. The square shutter partially occluding the food hopper opening was introduced to prevent long-billed or persistent birds stealing food when the hopper was in the lowered position—a problem encountered during early testing. In a subsequent modification, this problem was eliminated by introducing a horizontal moving shutter that completely blocked access to the food when the hopper lowered. The illuminable pecking key was utilised in experiment 4 only.

Body mass

In all experiments, body masses were recorded by the SFS each day between lights-on and lights-off; each mass was recorded with a bird identity corresponding to the microchip of the bird on the perch. The balances measured masses at a frequency of 6 Hz. A stable mass was recorded for a bird if the balance measured five consecutive masses of >50 g that were within a range of 5 g. These criteria were chosen to eliminate masses from birds that were perching incorrectly (e.g., by placing one foot on the food hopper), but to maximise the number of stable masses recorded from moving birds. Once a stable mass had been recorded another stable mass could not be recorded until the balance had measured a mass <10 g indicating that the current bird had left the perch. Balances were checked with a 100-g test mass a minimum of twice daily and calibrated if necessary. In order to control for build-up of guano on the perch over the day, balances were automatically tared regularly throughout the day when no bird was present on the perch.

The SFSs recorded a mean of 4.64 stable masses per bird per daylight hour over the four experiments. The raw masses showed a clear trend of mass increase over the day as expected (Fig. S1), but there was a lot of random error due to the imprecision of the balances and movement of the birds whilst on the perch, and masses were not always available at all times of every day. To estimate comparable dawn and dusk masses for each bird on each day we used the following procedure to model the available data. As long as a minimum of 10 masses were available, the mass data from each bird on each day were fitted with a cubic polynomial (Fig. S1A). To remove biologically impossible outliers, any masses >3 g from the fitted line were removed and a new cubic polynomial was refitted to the remaining data. This latter fit was used to estimate dawn and dusk masses for that day. To avoid extrapolation beyond the data, a dawn or dusk mass was only estimated if there was a data point within 1 h of the estimate. The above procedure was devised based on a detailed exploration of the data from experiment 4 and then applied unaltered in experiments 1–3. The one exception being that due to differences in daylight hours between experiments, dawn mass was defined as the fitted mass at 0900 for experiments 1–3 and 0615 for experiment 4; dusk mass was defined as the fitted mass at 1800 for experiments 1–3 and 1815 for experiment 4 (Fig. S1B). Thus, the dusk masses for experiments 1–3 and experiment 4 were estimated 9 and 12 h after lights-on respectively. A later time was not chosen for the dusk masses in experiment 4 due to the fact that birds often stopped foraging considerably before lights-off.

Operant foraging behaviour

In experiment 4, key pecks were recorded by the SFS between 0615 and 2045 each day. The time of each key peck was recorded with a bird identity and whether or not the peck was reinforced with food (probability of 1.0 under food security and either 0.4 or 0.2 under food insecurity for FIlow and FIhigh respectively).

Food consumption

Total food consumption in each aviary was estimated daily in all experiments by calculating the difference in the mass of the SFS food hoppers at the beginning and end of the day and subtracting any food collected in a spill tray located beneath each hopper. Hence, consumption data were only available at the aviary level.

Energy density of guano

Energy density of guano was measured by bomb calorimetry in experiments 1 and 3. Guano samples were collected daily from plastic trays positioned beneath perches in each aviary avoiding any feathers or wood chippings. Hence, guano data were only available at the aviary level. Samples were immediately frozen at −80 °C for storage. On completion of the experiment, samples were dried in a flow oven at 55 °C for 48 h until stable masses were obtained and finely ground to homogenise. In experiment 1, samples from each aviary were pooled over three consecutive days (yielding pooled samples centred on days 2 and 6 of FS1 and FS2 and days 2, 6, 9 and 13 of FI), whereas in experiment 3 all days were measured separately. For both experiments, each pooled sample was divided into two technical replicates prior to analysis. Bomb calorimetry was outsourced to Pemberton Analytical Services, Shropshire, UK and was conducted blind to the treatment group and technical replicates using a PAR 1261 adiabatic bomb calorimeter with a stated precision of ±0.1% on two determinations.

Physical inactivity

Data on physical inactivity were collected in experiments 1–3. As a metric of daytime physical inactivity we measured roosting behaviour—defined as perching motionless on a high rope perch. Video recordings of the aviaries were made for 15 min starting at 0900 and 1000 every day, before any food insecurity started, using a remotely operated wide-angle surveillance camera mounted in the ceiling of each aviary. The videos were scored manually using BORIS video coding software (Friard & Gamba, 2016). Scoring of roosting was blind to the experimental treatment in place on the previous day (the relevant predictor variable). A scan sampling method was used, whereby the instantaneous behaviour of each bird in the aviary was scored every 30 s for each 15-minute video. Birds were not individually identifiable on the videos, therefore the behavioural metric was the proportion of birds in the aviary observed roosting on each scan.

Statistical analysis

Data were analysed using R version 3.5.1 (R Core Team, 2018). Below we provide an overview of our statistical approach. Further details of statistical models are given in the Tables S2–S9.

For analysis of the effects of food insecurity in individual experiments we used linear mixed models (GLMMs) fitted using the R package ‘lme4’ (Bates et al., 2015). The models contained random effects (intercepts) to account for sources of non-independence in the datasets. In all models we analysed the effect of treatment: FS1 vs FI vs FS2 in experiments 1 and 3; FS1 vs FI in experiment 2; and FS1 vs FIlow vs FIhigh in experiment 4. In cases where a treatment continued for two weeks, the data were pooled from the first and second weeks of a treatment for statistical analysis, but the graphs in Fig. 3 show the data broken down by week. Since roosting data were proportions, they were arcsine square root transformed prior to analysis. Following any necessary transformation, all models gave satisfactory distribution of residuals, hence a Gaussian error structure was assumed throughout. We used restricted maximum likelihood estimation (REML) and conducted overall significance tests of treatment in the GLMMs using Satterthwaite’s method.

Figure 3 Effects of food insecurity on body mass, food consumption and energetic efficiency in experiments 1–4.

(A–D) Dawn mass (g); (E–H) dusk mass (g); (I–L) total daily food consumption per bird (g); and (M–P)energetic efficiency, calculated as the ratio of the mean dawn mass for a bird on a given day to the total mass of food eaten per bird on the previous day. Graphs are box plots with each box corresponding to 7 days of data, but the data from treatments lasting 14 days were pooled for statistical analysis. Mass data were available at the individual bird level, but consumption (and hence also efficiency data) were only available at the aviary level. For this figure, the three separate aviaries are combined for experiments 1 and 4. For display purposes only, data were within-subject centred and shown relative to the grand mean. All experiments involved 6 birds. Experiments 1–3 were run consecutively with the same six males (with breaks of 24 and 18 days between experiments), whereas experiment 4 was run with six females. Significance tests are presented in Tables S2, S3, S6 and S7; * p < 0.05, ** p < 0.01, *** p < 0.001.

For meta-analysis we used multi-level random effects meta-analysis models fitted using the R package ‘metafor’ (Viechtbauer, 2010). The models contained random effects of aviary nested in experiment. The effect sizes used in the meta-analyses were obtained from GLMMs fitted to the data from each aviary, using the same models for the analysis of individual experiments described above (with the exception that random effects of aviary were no longer required). Effects of food-insecurity were the parameter estimates (β values) and associated standard errors corresponding to: the difference between FS1 and FI for experiments 1–3 and the difference between FS1 and FIhigh for experiment 4. Tests of modifiers were conducted using meta-regression, whereby both baseline mass and competition (number of birds per SFS station) were added to the meta-analytic models. Estimation was by REML in all meta-analytic models. We used aviary-level data as the unit of analysis for the meta-analyses, because in the experiments involving multiple aviaries (1 and 4) there was considerable heterogeneity in results obtained from the different aviaries. Furthermore, increasing the number of replicates available for the meta-analysis increased the power available for the meta-regressions, making it possible to test for effects of baseline mass and competition.

Ethical statement

The study adhered to ASAB/ABS guidelines for the use of animals in research. Birds were taken from the wild under Natural England permit 20121066 and the research was completed under UK Home Office licence PPL 70/8089 with approval of the Animal Welfare and Ethical Review Body at Newcastle University. Prior to each individual experiment, a detailed protocol was approved by Newcastle University’s Comparative Biology Centre, but these protocols were not publicly pre-registered.

Results

Description of the dataset

Data were collected on 12 starlings comprising 6 males used in experiments 1–3 and 6 females used in experiment 4. Data were analysed at the individual bird level for body mass and key pecking, and at the aviary level for food consumption, energetic efficiency, guano energy density and roosting behaviour. In total, the experimental treatments ran for a total of 91 days in 8 separate aviaries across the four experiments. Data were collected on 210 aviary days and 546 bird days. On a few days, data were lost for one or more outcome variables due to experimenter error, equipment failure or insufficient mass data for the fitting algorithm (see Methods). Details of the unit of analysis and final sample size for each analysis are provided in Table S2–S8.

Body mass

Two measures of body mass were estimated for each bird on each day of each experiment: dawn mass, which is the mass before any foraging commenced and reflects changes in body composition including fat reserves, and dusk mass, which is the mass after a day of foraging and thus additionally includes gut content. To investigate the effects of food insecurity in each experiment, mixed-effects general linear models (GLMMs) were used to compare body mass on all days during the initial period of food security (FS1) with body mass on all days during the subsequent period of food insecurity (FI in exp. 1–3 and FIhigh in exp. 4; Figs. 3A–3H and Fig. S2; Tables S2 and S3). To summarise the effects from the four experiments combined, we used random effects (RE) meta-analysis. Overall, dawn mass increased by 0.52 g under FI (95% CI [−0.12 to 1.17]), but this change was not significant (RE meta-analysis: z = 1.59, p = 0.113; Fig. 4A; Table S4). Dusk mass increased by 1.51 g under FI (95% CI [0.78–2.25]; RE meta-analysis: z = 4.05, p < 0.001; Fig. 4B; Table S4). The increase in dusk mass corresponds to ∼3% of the mean dusk mass during FS1.

Figure 4 Meta-analysis of the effects of food insecurity in experiments 1–4.

(A) Dawn mass (g); (B) dusk mass (g); (C) total daily food consumption per bird (g); and (D) energetic efficiency, calculated as the ratio of the mean dawn mass of a bird on a given day to the total mass of food eaten per bird on the previous day. Graphs are forest plots indicating the size of effects and 95% CI for the difference between the first period of food security (FS1) and the period of food insecurity (FI or FIhigh). We treated each aviary in experiments 1 and 4 as separate replicates due to heterogeneity between aviaries. Meta-analyses used a multi-level random-effects model with random effects of aviary nested in experiment. All experiments involved six birds. Statistical tests are presented in Table S4.

The meta-analysis showed heterogeneity among the effects of food insecurity on dawn and dusk mass across the four experiments (tests for heterogeneity: τ2 ≥ 0.95, Q7 ≥ 29.79, p < 0.001; Table S4), suggesting variation in how birds responded to FI. Two possible moderators of the effect of food insecurity on mass gain were explored: the baseline mass of the birds during FS1 (mean dawn mass) and the degree of competition in the aviary (low, where 2 birds shared one SFS station, or high, where 6 birds shared two SFS stations; see Fig. 1). Addition of these two moderators to the meta-analytic model explained ∼82% of the heterogeneity in the effects of FI on dawn mass (omnibus test of meta-regression: QM2 = 18.52, p < 0.001) and the test for residual heterogeneity was not significant (QE5 = 5.56, p = 0.352). The FI-induced increase in dawn mass was greater for aviaries with lower baseline body mass (βbaseline mass = −0.45 95% CI [−0.67 to −0.23], z = −4.06, p < 0.001; Fig. 5) and for aviaries with higher competition for access to food (βhigh competition = 1.76 95% CI [0.55 to 2.96], z = 2.86, p = 0.004; Fig. 5). The results for dusk mass were similar. The moderators explained ∼76% of the heterogeneity in the effects on dusk mass (omnibus test of meta-regression: QM2 = 16.34, p < 0.001). The effect of FI on dusk mass gain was greater for aviaries with lower baseline body mass (βbaseline mass = −0.50 95% CI [−0.67 to −0.23], z = −2.90, p = 0.004) and in aviaries with higher competition (βhigh competition = 3.60 95% CI [1.76 to 5.44], z = 3.84, p = 0.001). However, the test for residual heterogeneity was still significant in this case (QE5 = 12.63, p = 0.027). Note that the moderating effect of baseline mass is not an artefact of measurement error and regression to the mean, because the mass estimates are based on multiple measurements and are therefore highly precise.

Figure 5 The moderating effect of baseline mass on the effect of food insecurity on body mass.

Baseline mass is the mean dawn mass (g) in the first period of food security (FS1). The effect of FI is the parameter estimate (b) representing the difference in dawn mass between the first period of food security (FS1) and the period of food insecurity (FI or FIhigh) taken from mixed linear model for each aviary of each experiment. Points are plotted with size inversely proportional to the precision of the estimates. The horizontal dotted line shows no effect of food insecurity on dawn mass. The fitted lines come from a multi-level meta-regression model with baseline mass and degree of competition as moderators and random effects of aviary nested in experiment; the solid line is the fit for low competition and the dashed line the fit for high competition.

To investigate the effect of reinstating food security in experiments 1 and 3, dawn and dusk mass in FS2 were compared with those in FS1 and FI. Reinstating food security had no effect on dawn mass in either experiment (Figs. 3A, 3C; Table S2), but dusk mass declined in both experiments relative to FI back to a level not significantly different from that in FS1 (Figs. 3E, 3G; Table S3).

Foraging behaviour

Key pecking data from the operant schedules in experiment 4 were used to investigate how birds responded to the reduced probabilities of reinforcement in FIlow and FIhigh. Birds increased their frequency of key pecking during both periods of FI compared to FS1 (GLMM: F2,118 = 87.16, p < 0.001; Fig. 6A, Table S5). The birds also exploited each 10-s reinforcement more effectively, consuming more food per second in FI (GLMM: F2,55 = 23.07, p < 0.001; Fig. 6B, Table S5). These responses were graded, with birds pecking and eating faster under FIhigh than under FIlow (Table S5). However, this increased foraging effort was insufficient to compensate for the reduced probabilities of reinforcement in FI. The birds earned fewer reinforcements per day (GLMM: F2,118 = 32.95, p < 0.001; Table S5) and overall ate less food per day in FI (GLMM: F2,54 = 8.77, p < 0.001; Fig. 3L; Table S6).

Figure 6 The effect of food insecurity on foraging behaviour in experiment 4.

A) Peck rate (pecks hr−1); and (B) consumption rate (g food eaten reinforcement−1). Graphs are box plots of data from experiment 4, with each box corresponding to 7 days of data. The green diamonds in A indicate the peck rate that would have been required to fully compensate for the reduced probability of reinforcement in the food-insecure treatments and hence receive the same rate of reinforcement obtained under food security (black diamond). For display purposes, data were within-subject centred and plotted relative to the grand mean. The n was 6 birds. Significance tests are presented in Table S5. *** p < 0.001.

Food consumption

The mean consumption per bird each day was computed (by dividing the measured consumption in an aviary by the number of birds present) and compared on all days during FS1 with all days during FI for each experiment (Figs. 3I–3L; Table S6). Overall, food consumption dropped by 2.50 g bird−1 day−1 (95% CI: [−3.13 to −1.87]) when birds moved from FS1 to FI (RE meta-analysis: z = −3.08, p = 0.002; Fig. 4C; Table S4). This corresponds to a ∼13% drop in daily consumption compared to the initial period of food security. There was heterogeneity among the effects of FI on food consumption (tests for heterogeneity among associations: τ2 = 3.20, Q7 = 61.76, p < 0.001; Table S3). Addition of baseline body mass and competition to the meta-analytic model as moderators did not explain a significant percentage of this heterogeneity (omnibus test of meta-regression: QM2 = 5.76, p = 0.056), although given the marginal nature of this result there is potentially some evidence for an effect here.

Reinstating food security had no effect on consumption in experiment 1, but caused an immediate increase in consumption in experiment 3 (Fig. 3K; Table S6). However, consumption during FS2 remained lower than during FS1 in both experiments (Figs. 3I, 3K; Table S6).

Energetic efficiency

We calculated energetic efficiency as the body mass maintained per mass of food consumed, using dawn mass and the amount of food consumed on the previous day. We compared all days during FS1 against all days during FI for all 4 experiments (Figs. 3M–3P, Table S7). Overall, energetic efficiency increased by 0.85 (95% CI [0.65 to 1.06]) when birds moved from FS1 to FI (RE meta-analysis: z = 8.17, p < 0.001; Fig. 4D, Table S4). This corresponds to a ∼18% increase in energetic efficiency compared to FS1. The effect of FI on energetic efficiency (1.15 sd) was larger than that on dawn or dusk mass alone (0.17 and 0.42 sd respectively). Furthermore, the effect of FI on energetic efficiency was less heterogeneous between experiments and aviaries than the effects on body mass, although the heterogeneity was still significant (τ2 = 0.20, Q7 = 29.35, p < 0.001). Addition of baseline body mass and competition to the meta-analytic model as moderators did not explain a significant percentage of the heterogeneity (omnibus test of meta-regression: QM2 = 5.26, p = 0.072).

Reinstating food security had no effect on efficiency in experiment 1, but caused an immediate decrease in efficiency in experiment 3 (Figs. 3M, 3O; Table S7). However, efficiency during FS2 remained higher than FS1 in both experiments (Figs. 3M, 3O; Table S7).

Energy density of guano

In experiments 1 and 3 bomb calorimetry was used to measure the effect of FI on the energy density of homogenised pooled samples of guano that were collected to be representative of all birds in an aviary. Energy density was compared on all days during FS1 with all days during FI for each experiment (Fig. 7, Table S8). Overall, energy density dropped by 0.12 MJ. kg−1 (95% CI: [−0.22 to −0.01]) when birds moved from food security to food insecurity (RE meta-analysis: z = −2.18, p = 0.030; heterogeneity: τ2 = 0, Q3 = 1.50, p = 0.681; Fig. 7C). This corresponds to a ∼1% drop in energy density compared to FS1. In experiment 1, in which food insecurity lasted for 14 days, the energy density of guano decreased during FI relative to FS1 (βFI = −0.17 95% CI [−0.33–−0.01], t-test, t19 = −2.12, p = 0.047), whereas in experiment 3, in which FI lasted for only 7 days, there was no difference between FI and FS1 (βFI = −0.07 95% CI [−0.22 to 0.09], t-test, t53 = −0.84, p = 0.403).

Figure 7 The effect of food insecurity on energy density of guano.

(A) Experiment 1; and (B) experiment 3. Graphs show boxplots of the energy density of guano with each box corresponding to 7 days of the experiment. For experiment 1, data were within-aviary centred and plotted relative to the grand mean. Significance tests for A and B are presented in Table S8: * p < 0.05, ** p < 0.01. (C) Forest plot of meta-analysis of the pairwise comparison between FS1 and FI in experiments 1 and 3.

Reinstating food security had no effect on energy density of guano in experiment 1 (βFS2 = −0.06 95% CI [−0.22 to −0.10], t-test, t19 = −0.77, p = 0.452), whereas it was associated with a decrease in energy density in experiment 3 (βFS2 = −0.17 95% CI [−0.31 to −0.03], t-test, t53 = −2.45, p = 0.018). In both experiments, energy density was lower during FS2 than FS1 (exp. 1: βFS2 = −0.23 95% CI [−0.41 to −0.05], t-test, t19 = −2.51, p = 0.022; exp 3: βFS2 = −0.24 95% CI [−0.37 to −0.10], t-test, t53 = −3.42, p = 0.001).

Roosting behaviour

In experiments 1–3 the proportion birds that were observed roosting in instantaneous scans of the aviary made every 30 s was scored from video as a metric of physical inactivity. Scans were made during the period between 0900 and 1100 when ad libitum food was always available regardless of the treatment in place that day. We modelled the effects of the treatment experienced the previous day on roosting (Figs. 8A–8C; Table S9). Overall, the proportion of birds observed roosting did not change significantly under FI (RE meta-analysis on arcsine square root-transformed proportions: estimate = 0.03 95% CI [−0.14 to 0.08], z = −0.46, p = 0.645; heterogeneity: τ2 = 0.04, Q4 = 33.59, p < 0.001; Fig. 8D). In experiment 3 roosting increased significantly under FI (βFI = 0.17, 95% CI [0.08 to 0.27]; t-test, t24 = 3.47, p = 0.002), but in aviary 1 of experiment 1 the effect was in the opposite direction.

Figure 8 The effect of food insecurity on roosting behaviour.

(A) Experiment 1; (B) experiment 2; and (C) experiment 3. Graphs show boxplots of the proportion of scans in which birds were roosting with each box corresponding to 7 days of the experiment. For experiment 1, data were within-aviary centred and plotted relative to the grand mean. Significance tests for panels A–C are presented in Table S9: * p < 0.05, ** p < 0.01, *** p < 0.001. (D) forest plot of meta-analysis of the experiments 1–3.

Reinstating food security was associated with an increase in roosting in experiment 1 (βFS2 = 0.27 95% CI [0.16–0.38], t-test, t76 = 4.87, p < 0.001), but no change in roosting in experiment 3 (βFS2 = −0.05, 95% CI −0.13 to 0.03; t-test, t23 = −1.19, p = 0.245). In both experiments, roosting was more common during FS2 than FS1 (exp. 1: βFS2 = 0.21 95% CI [0.09–0.34], t76 = 3.39, p = 0.001; exp 3.: βFS2 = 0.12 95% CI [0.04–0.20], t-test, t23 = 2.79, p = 0.011).

Discussion

Across four experiments, starlings responded to food insecurity by increasing their dusk body mass by ∼3%. Moreover, food insecurity caused increased body mass in experiment 4 where there were no periods of food deprivation lasting more than a few minutes, indicating a response to a relatively subtle increase in the short-term unpredictability of food availability within a day. In none of the four experiments did the food-insecure birds consume more food in total. In direct opposition to the predictions of recent mechanistic models (Anselme, Otto & Güntürkün, 2017; Anselme & Güntürkün, 2019), food insecurity was associated with a ∼13% decrease in daily food consumption. Our data therefore lead to the novel conclusion that starlings respond to food insecurity by increasing their energetic efficiency—the body mass maintained per unit of food consumed per day—by ∼18%. Although food-insecure birds gained dusk mass on average, increasing energetic efficiency was the more robust response, characterised by larger effect sizes and less variability in response. The increase in energetic efficiency observed under food insecurity may be partially explained by the birds assimilating more energy from their food, and in experiment 3, by reducing energy spent on physical activity. The changes caused by food insecurity did not immediately reverse when food security was reinstated: energetic efficiency, energy absorption and physical inactivity all remained higher than they had been at baseline for 1–2 weeks following reinstatement of food security.

Although food insecurity caused an increase in body mass overall, there was variation both within and between birds in the response: some birds maintained their baseline mass, while others gained mass and the overall effect of food insecurity varied between experiments. Our meta-regression results shed some light on the possible causes of this variation. In support of previous results from birds, we showed that starlings were more likely to gain mass under food insecurity if they were lighter at baseline (Pravosudov & Grubb, 1997; Witter & Swaddle, 1997) and if they faced greater competition for food in the aviary (Witter & Swaddle, 1995; Witter & Goldsmith, 1997). Both of these findings make adaptive sense. If increased mass provides insurance against starvation, then thinner birds should obtain greater benefits from mass gain than birds who already have sufficient fat reserves to survive periods without food. Birds facing higher competition effectively face harsher food insecurity than birds facing lower competition, increasing the risk of starvation and hence the fat reserves it is optimal to carry. The finding that both baseline mass and competition moderate the effect of food insecurity on body mass potentially helps to explain why published studies of the effects of unpredictable food do not always report mass gain.

Increasing energetic efficiency implies either increasing the amount of energy absorbed from food, or decreasing energy expenditure in some domain. We found suggestive evidence for the former strategy and inconsistent evidence for the latter. The energy density of the birds’ guano decreased under food insecurity, suggesting that the birds were assimilating more energy. Increased energy assimilation under reduced food intake has previously been reported in starlings (Bautista et al., 1998). Moreover, rats and rhesus macaques subject to long-term caloric restriction paradigms do not reduce their total daily energy expenditure by the amount predicted by their reduced food intake, suggesting that these species too must increase their energy assimilation when intake is restricted (Selman et al., 2005). Since energy assimilated is likely to be a decelerating function of gut residence time, this result could simply be a passive physical consequence of increased gut passage time resulting from slower food intake. Another (not mutually exclusive) explanation, is that the birds responded to food insecurity by strategically changing their gut anatomy or physiology to increase energy assimilation. Starlings are able to adaptively alter their gut morphology in response to changes in diet (Al-Joborae, 1979; Geluso & Hayes, 1999), demonstrating that gut plasticity in response to food insecurity is not implausible. Given that the change in energy density of guano that we observed was slow, reducing over the 2-week period of food insecurity in experiment 1, and did not appear to reverse immediately when food security was reinstated, our data favour a strategic adaptation. Therefore, we hypothesise that starlings respond to food insecurity by altering their gut in some way to increase energy assimilation. However, it seems unlikely that increased energy assimilation could be the only explanation for increased energetic efficiency in the current dataset, due to the relatively small size of the effect (∼1% drop in energy density of guano) compared with the large decrease in food consumption (∼13% drop in daily food consumption).

In four of five aviaries where we measured physical inactivity, roosting behaviour increased under food insecurity. An overall effect of food insecurity was rendered null by one aviary where the effect went strongly in the opposite direction. Roosting occurs when the birds are not engaged in other activities such as foraging, eating or bathing, and is likely to be associated with the lowest levels of energy expenditure due to physical activity. We measured roosting during the first two hours of the day when food was always available ad libitum, meaning that any changes in roosting were not a direct response to the current unavailability of food. The fact that the increase in roosting observed in experiment 3 was simultaneous with body mass gain, suggests that reduced physical activity could have been causal in mass gain. However, since food insecurity did not induce a significant increase in roosting in either experiments 1 or 2, it seems unlikely that decreased physical activity is sufficient to explain the increases in energetic efficiency observed in all experiments, even in conjunction with increased energy assimilation. It is noteworthy that although zebra finches do not respond to unpredictable food deprivation by increasing body mass (Dall & Witter, 1998; Marasco et al., 2015), they do decrease physical activity (Dall & Witter, 1998), compatible with an increase in energetic efficiency.

We did not measure metabolic rate in the current study, but reducing basal metabolic rate could be a third mechanism by which food-insecure birds increased energetic efficiency (Wiersma, 2005; Secor & Carey, 2016). Food insecurity might cause the birds to down-regulate or turn off energetically expensive, but temporarily expendable, biological systems in order to maximise probability of survival in the face of short-term energetic shortfalls. Candidate systems include, somatic maintenance, the immune system and the reproductive system. The costs associated with increased energetic efficiency might therefore be measured in terms of accelerated biological ageing, increased risk of cancer and infectious disease, or reduced reproductive success. No relevant data were collected in the current study, but previous studies in passerine birds have provided evidence for the existence of such trade-offs (e.g., Wiersma & Verhulst, 2005; Lynn et al., 2010; Cornelius et al., 2017).

Limitations and strengths

Each of the experiments in this study used only six birds. Small sample sizes simultaneously increase the probability of false negative results and the probability that any positive results obtained are false positives (Bateson & Martin, 2021). The probability of obtaining false negative results will have been further exacerbated by our decision to use all days of data from each treatment in the analyses presented. Since birds adjusted to changes in treatment over a period of days (see Fig. S2 for an example from the body mass data), this was a conservative decision, resulting in a likely underestimation of treatment effects that will have been greatest when treatments lasted for only 7 days. While this decision is likely to have contributed to false negative results in some individual experiments, it was made to avoid the complexity and added researcher degrees of freedom that would have been introduced by identifying and implementing stability criteria for each of the output variables within each treatment.

To mitigate against false negatives, we used longitudinal within-subjects designs that increased our power to detect treatment effects. We measured individual body mass in unprecedented detail, collecting a mean of 4.64 stable masses per bird per daylight hour across the four experiments. These data permitted modelling of individual within-day trajectories in body mass, reducing the impact of measurement error and increasing confidence in mass estimates. Finally, our headline conclusions are based on summary estimates of effect sizes obtained from meta-analysis of the results obtained from multiple experiments (some of which were individually inconclusive). Since our conclusions are positive findings, concerns about false negatives are irrelevant.

Could our headline results be false positives? The probability that a positive finding is a true positive is known as the positive predictive value (PPV). All else being equal small studies have lower PPV, but PPV is increased by the probability that a given finding was true before the study was conducted (Button et al., 2013). Given the prior evidence that unpredictable food should and does increase body mass in starlings and other species, the result that food-insecurity increased dusk mass is likely to be true. In contrast, the result that food insecurity reduced food consumption and hence increased energetic efficiency was not predicted beforehand. Therefore, despite our significant meta-analytic findings, these results have lower PPV and should be treated with greater scepticism. However, there are reasons for confidence in these findings, as we explore below.

The experiments differed in a number of methodological details including the age, sex and previous experience of the birds, the number of birds in an aviary, the competition for feeder access, the method used to induce food insecurity and the number of weeks for which each treatment was maintained. This heterogeneity undoubtedly contributed to variability in the results and might be seen as a limitation of the dataset. However, there is a trade-off between standardisation to reduce variability and heterogenisation to increase the generality of findings. Over-standardisation of protocols for animal experiments has been highlighted as a likely cause of replication failure (Voelkl & Würbel, 2016). Triangulation of conclusions from multiple different experiments, as we have done in the current study, is recognised as a powerful strategy for reducing false-positive findings due either to chance, or caused by a specific confounding variable or moderator present in a single experiment (Munafò & Smith, 2018).

Due to constraints of our methodology, food consumption (and hence energetic efficiency), energy density of guano and roosting behaviour were only available at the aviary level. This is undoubtedly a limitation, because aviary-level effects could have been due to effects in only a subset of the birds present. This concern is somewhat mitigated by the fact that the effects of food insecurity on food consumption, energetic efficiency and energy density of guano all emerged from meta-analytic summaries of multiple experiments.

Obtaining accurate individual data for variables such as food consumption is currently only possible in individually housed birds. Due to space constraints, this usually implies housing birds in smaller cages that restrict natural behaviour (and compromise animal welfare). Our aim in this study was to create a refined and more ecologically valid model of body mass regulation in which birds could interact socially, walk, fly, water bathe and forage more naturally. Many of these behaviours are either impossible or severely limited in individual cages, meaning that caged birds have less latitude to modify their energy expenditure via changes in physical activity. Therefore, there is a trade-off between the level at which some variables can be measured and the ecological validity of experiments that has implications for the types of effects that are likely to be detected. Given the established importance of the social and physical environment for body mass (Miles et al., 2009), we believe that a move towards more naturalistic experimental paradigms in which opportunities for social interaction and physical exercise are present is critical, even if this comes at the cost of limiting individual-level measurement of some variables.

A final limitation is that our analyses of guano energy density were based on representative guano samples collected from the aviaries, rather than on the total amount of guano produced. In order to calculate the proportion of ingested energy that is lost in guano it would be necessary to know the relative amount of guano produced in the different treatments. Due to the size of the aviaries and the use of wood chips on the floor, accurately collecting total guano production was simply not feasible.

Implications for human obesity

Our findings highlight the dangers of extrapolating from short-term laboratory studies showing increased food consumption in food-insecure humans (Nettle et al., 2018; Stinson et al., 2018) to the assumption that energy intake is increased by food insecurity in the home environment. The results from experiment 4 show that it is possible for food-insecure birds to show increased foraging motivation, yet still consume less food overall in a food insecure environment. The lack of evidence for increased energy intake in food-insecure humans from analysis of 24-hour food recall data has been dismissed as under-reporting of food intake by the obese (Stinson et al., 2018). However, an alternative hypothesis suggested by our results is that mass gain in food-insecure humans is explained by reduced energy expenditure or increased assimilation of energy from food, rather than increased intake.

Only women show increased body mass associated with food insecurity (Nettle, Andrews & Bateson, 2017), meaning that much research on human food-insecurity has been restricted to women. However, our results raise the question of whether increased energetic efficiency has been missed in men. A longitudinal study showing that girls responded to becoming food insecure by gaining body mass, despite no change in energy intake, also reported that boys responded by maintaining their body mass, despite reducing their energy intake (Jansen et al., 2017). These results support a common effect of food insecurity on energetic efficiency in human children of both sexes. Further work is needed to establish whether this result is also found in food-insecure adults.

As a final note, we have encountered some scepticism over whether passerine birds are appropriate animal models for understanding human strategic mass regulation (e.g., Speakman, 2018). Critics focus on two related arguments concerning the relevance of the insurance hypothesis (which was originally developed for birds) to humans. The first is that the starvation-predation trade-off assumed to underpin strategic mass regulation in birds does not apply to humans, because predation has not been an important selection pressure in our recent evolutionary history. The second argument is that the costs of excess body mass present in birds do not apply in humans, because humans do not fly. In response to these arguments we emphasise that the insurance hypothesis does not rely on either the existence of a starvation-predation trade-off underpinned by mass-dependent predation, or the costs of flight. The theory simply requires that there is an asymmetrical inverted U-shaped relationship between reserves and Darwinian fitness, such that fitness declines rapidly as reserves decrease below their optimum level and less rapidly as reserves increase above their optimal level (Nettle, Andrews & Bateson, 2017). Humans may not fly, but we are adapted to walk and run long distances (Pontzer, 2017), suggesting that endurance locomotion was important for fitness during our recent evolutionary history. Furthermore, human athletic performance in distance events declines continuously with higher body mass (Sedeaud et al., 2014). Therefore, the costs of being overweight will be incurred in any human activity involving endurance locomotion, whether this is fleeing predators, foraging for food, fighting wars or attracting mates.

Conclusions

Using a novel, more naturalistic system for studying foraging and mass regulation in starlings we have demonstrated that while birds gained body mass under food insecurity, replicating many previous findings, they achieved this despite eating less. It is therefore clear that mass gain under food insecurity is not mediated by increased food consumption in this system, refuting a common assumption in the literature. Furthermore, our results suggest that a more general effect of food insecurity is to induce increased energetic efficiency. Further work is required to understand the mechanisms underlying this effect in starlings and to explore whether the results generalise to other species including humans.

Supplemental Information

Supplemental Information 1 ARRIVE guidelines 2.0

Author checklist.

Click here for additional data file.

Supplemental Information 2 Raw body mass data and method for estimating dawn and dusk masses

(A) Scatterplot of body mass against time of day for a single bird on a single day in experiment 4. The red line shows the polynomial fit to the raw data. The dashed black lines show the 3 g band either side of the fitted line; masses lying outside this band were deleted. (B) Scatter plot of the cleaned body mass data. The red line shows the slightly modified polynomial fit after removing the outliers. The vertical dotted lines show the times designated as dawn and dusk in experiment 4 (0615 and 1815 respectively) and the blue arrows show the estimates of dawn and dusk mass for this bird on this day.

Click here for additional data file.

Supplemental Information 3 The rapidity of the effect of food insecurity on body mass

(A) Dawn mass (g) and (B) dusk mass (g). Graphs are box plots of the mass data from experiment 2, with each box corresponding to a single day of data. For dawn mass the treatment is that in place the previous day, whereas for dusk mass the treatment is that in place the same day (the first box of panel A is white because there was no treatment in place the previous day). The vertical lines indicate when treatment changed from FS1 to FI. For display purposes, data were within-subject centred and plotted relative to the grand mean. The n was 6 birds.

Click here for additional data file.

Supplemental Information 4 Details of the four experiments

Click here for additional data file.

Supplemental Information 5 Summary of linear mixed models of effects of food insecurity on dawn1 mass (g)

Click here for additional data file.

Supplemental Information 6 Summary of linear mixed models of effects of food insecurity on dusk1 mass (g)

Click here for additional data file.

Supplemental Information 7 Summary of meta-analysis1 of body mass, food consumption and energetic efficiency in experiments 1–4

Click here for additional data file.

Supplemental Information 8 Summary of linear mixed models of effects of food insecurity on foraging variables in experiment 4

Click here for additional data file.

Supplemental Information 9 Summary of statistical analyses of effects of food insecurity on total food consumption (g/bird/day)1

Click here for additional data file.

Supplemental Information 10 Summary of linear mixed models of effects of food insecurity on energetic efficiency

Click here for additional data file.

Supplemental Information 11 Summary of linear mixed models of effects of food insecurity on energy density of guano (MJ/kg)

Click here for additional data file.

Supplemental Information 12 Summary of linear mixed models of effects of food insecurity on the proportion of scans in which a bird was roosting

Click here for additional data file.

We thank: Greg Prescott and his team of engineers at Campden Instruments Ltd. for working with us to develop the SFS; Rudolph Cardinal for programming assistance; Michelle Waddle and her team of technicians in the Comparative Biology Centre at Newcastle University for caring for our birds; Richard Bevan for use of his oven; Marie-Claire Pagano for assistance with experiments; all members of the COMSTAR group and Centre for Behaviour and Evolution for helpful discussion; and Gillian Pepper for comments on the manuscript.

Additional Information and Declarations

Competing Interests

Author Contributions

Animal Ethics

Field Study Permissions

Data Availability

The authors declare there are no competing interests.

Melissa Bateson conceived and designed the experiments, performed the experiments, analyzed the data, prepared figures and/or tables, authored or reviewed drafts of the paper, and approved the final draft.

Clare Andrews and Charlotte B.C.M. Egger performed the experiments, authored or reviewed drafts of the paper, and approved the final draft.

Jonathon Dunn and Francesca Gray conceived and designed the experiments, performed the experiments, analyzed the data, authored or reviewed drafts of the paper, and approved the final draft.

Molly Mchugh performed the experiments, analyzed the data, authored or reviewed drafts of the paper, and approved the final draft.

Daniel Nettle conceived and designed the experiments, performed the experiments, authored or reviewed drafts of the paper, and approved the final draft.

The following information was supplied relating to ethical approvals (i.e., approving body and any reference numbers):

Research was completed under UK Home Office licence PPL 70/8089 with approval of the Animal Welfare and Ethical Review Body at Newcastle University.

The following information was supplied relating to field study approvals (i.e., approving body and any reference numbers):

Birds were taken from the wild under a permit from Natural England permit 20121066.

The following information was supplied regarding data availability:

The datasets and R scripts are available at Zenodo: Bateson, Melissa, Andrews, Clare, Dunn, Jonathon, Egger, Charlotte, Gray, Francesca, McHugh, Molly, & Nettle, Daniel. (2019). Food insecurity increases energetic efficiency, not food consumption: an exploratory study in European starlings (Version Version 3) [Data set]. Zenodo. http://doi.org/10.5281/zenodo.4584627.

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
