# Peer review of "Food insecurity increases energetic efficiency, not food consumption: an exploratory study in European starlings"

_PeerJ, doi:10.7717/peerj.11541_

## Round 0.1 · original submission · Minor Revisions

In your revision please pay particular attention to the potential order effects raised by reviewer 2 and discussion of statistical significance discussed by reviewer 1.

·

Basic reporting

Excellent, bar a few small quibbles.

Experimental design

Excellent. It is acknowledged and discussed that each individual study is relatively small but meta-analyses are run across each experiment.

Validity of the findings

Robust, within the recognised limits of the sample sizes and the heterogeneity of the various experiments.

Additional comments

This report is a masterclass in undertaking, analysing and reporting on experimental studies. The experiments are clearly explained and complimentary, the limits to the study are clearly explained in an upfront and honest way and demonstrate strong stats literacy, and throughout the manuscript the authors explicitly recognise the limitations of the work based on sample size and various methodological constraints. One thing that possibly could be explored a little more, though I appreciate that this would somewhat go against the approach taken of looking across the studies together given that each individual study has low predictive power, is why there is such marked variation in outcome variables particularly between replicate cages in the same experiments, when this is the case. Finally, the links between humans and (other) animals adds an important additional dimension to the study.

Some minor comments and observations:
L49 – Surely there are studies quantifying food intake in people experiencing different levels of food insecurity.
L50 – Why past tense?
L118-122 – Here, I’m unclear what exactly the three experiments are (having re-read these sentences several times).
L134 – reinforcements of what, exactly?
L317, and indeed the rest of the paragraph – unclear. What is the variable of behaviour (with units) being used in this study? (It seems to be one variable? And about roosting?)… L332 – perhaps made clear here, but a bit late on.
L433 – The use of p = 0.05 for ‘significant or not significant’ is inherently problematic (as I appreciate you will know) and highlighted here. Would you have concluded that body mass and competition were important moderators here if e.g. p = 0.049? Play around with your data and you will find that just marginal tweaks to the measurement will create this amount of difference in p. Better to interpret this p value as ‘some evidence for an effect’?
Figure 4 shows some considerable variation in effect size (and direction) across the experiments, including across aviaries within experiments. For example, (A) experiment 1 aviary 3; (B) experiment 1 aviary 1. How different the findings would have been, and the conclusions drawn, had e.g. a single experimental type or single aviary been used. This is not a criticism, but shows the value of multiple replicates and experimental types, and then good, clear figures to show the results.

Reviewer 2 ·

Basic reporting

Excellent.

Experimental design

Potential order confound that needs explicitly discussing. Should not weaken main findings substantially.

Validity of the findings

Small sample sizes explicitly acknowledged. Findings are likely very robust to this.

Additional comments

Overall I really liked the work presented here. It addresses an important topic of broad interdisciplinary interest and generates novel, unexpected findings. Nevertheless I feel that the authors need to explicitly address a couple of issues to maximise the eventual impact of their work:

1. The authors discuss the potential importance of their work for understanding human obesity. However, I feel they do not acknowledge the limitations of using a small passerine in this context. In particular, since strategic body mass regulation is generally thought of as reflecting trade offs between the benefits of fat storage and the costs of carrying mass. In birds such costs are likely to be much more instantaneous than in terrestrial mammals (getting heavier will likely immediately impact flight performance) and so details of mass regulation dynamics are not likely to be easily generalisable to humans. I feel the authors need to acknowledge this explicitly in their discussion of the implications of their results for understanding obesity in humans.

2. As far as I can tell there is an unacknowledged order confound in their experimental design since all 6 male starlings they used went through experiments 1-3. We need details of the ordering of the experiments on these 6 birds (did they perform the experiments in numerical order? how long between experiments? etc). Since all 6 birds went through the three experiments together there is the potential for an effect of the sequence in the experimental series to impact their results. Although I can't see any way this would undermine their headline results, it needs to be acknowledged as it may help explain some of the differences among the experiments in the dynamics of the effects observed.

---

## Round 0.2 · accepted · Accept

Thank you for your updated version. I am happy to accept the manuscript based on the recommendation of the two reviewers.

·

Basic reporting

Good.

Experimental design

Good.

Validity of the findings

Good.

Additional comments

Great paper.

Reviewer 2 ·

Basic reporting

Fine

Experimental design

Fine

Validity of the findings

Fine

Additional comments

I'm fine with the revised MS now.

Nevertheless, I didn't appreciate the overly dismissive treatment of the issues I raised:

1. On the issue of using small birds as models - they attacked criticisms i didn't make. It's not that i don't believe that humans suffer mass-dependent costs (I'm sure they do). Rather, the dynamics of the marginal costs of being heavier are likely to differ fundamentally between flying and other forms of locomotion. For non-flyers mass-dependent costs are less instantaneous: being a little bit heavier is unlikely to impact your ability much to move efficiently/accurately if you are swimming/walking.. So the costs of carrying additional mass are likely to be deferred over longer time periods (you accumulate more damage, are less able to sustain long bouts of activity, you accumulate chronic physiological burdens associated with being fat). This will make the direct generalisation of insights from fine-scale dynamics of strategic body mass regulation from starlings to humans difficult. Despite missing this nuance in their rebuttal, the section they've added at least acknowledges there is a controversy and deals with the low-hanging fruit criticisms.

2. Just because none of us can think of an explanatory/independent variable that was impacted by the order in which the experiments were conducted doesn't mean there isn't some hidden/unmeasured carry over influence. As I understand it, this is included in discussions of "confounds" in experimental design. But the fact that they give the reader all the relevant info in the text now is sufficient.